# Enhancing Foreign Language Enjoyment through Online Cooperative Learning: A Longitudinal Study of EFL Learners

**DOI:** 10.3390/ijerph20010611

**Published:** 2022-12-29

**Authors:** Songyun Zheng, Xiang Zhou

**Affiliations:** College of Foreign Languages, Shanghai Maritime University, Shanghai 201306, China

**Keywords:** foreign language enjoyment, online cooperative learning, EFL learners, mental health, self-appraisal, reflection

## Abstract

This study examines university students’ foreign language enjoyment (FLE) in an online cooperative learning (CL) context and explores, taking a positive psychology approach, how and why CL may shape FLE. To this end, 98 Chinese university students studying English as a foreign language (EFL) were assigned into experimental (*n* = 49) and control groups (*n* = 49). Both groups completed a short-form foreign language enjoyment (FLE) scale before and after a 3-month intervention. The students in the experimental group were assigned with tasks that needed to be accomplished by teamwork. Moreover, each team was also requested to reflect upon their cooperation experiences and to self-assess their performance of these tasks. The results show that the overall FLE of the experimental group increased remarkably, whereas that of the control group fluctuated considerably. Furthermore, analyses of experimental group students’ self-appraisal comments revealed that students with pleasant cooperation experiences usually experience high FLE, give satisfactory marks on their performance, and feel confident about achieving better FL performance in the future. The findings and implications provide meaningful insights into how online FLE can be boosted through CL so as to promote positive mental health of students in a technology-assisted language learning (TALL) context.

## 1. Introduction

The “emotional deficit” in second language acquisition (SLA) research [1] has largely been made up by scholars’ keen attention paid to emotions. Anxiety, enjoyment, and boredom in particular have been examined closely by SLA researchers [2,3]. This reveals that emotions, which are ubiquitous, complicated, and fluctuating constantly in an educational context [4], are closely related to students’ motivation [5], performance [6], and well-being [7]. Parallel to the negative emotions endured by students who learn English as a foreign language (EFL), the research undertaken by scholars on their positive counterparts, such as enjoyment, has proven to be fruitful, as there is evidence that students can experience positive emotions if adequate teacher support is provided, a favorable learning environment is created, and real achievement is obtained [8]. Despite the increasing concentration on foreign language enjoyment (FLE), and apart from a handful of studies which adopted a longitudinal approach to investigate students’ emotions [5,9,10,11], little research with a mixed-method design has been carried out to examine how and why cooperative learning (CL) in an online context can shape university EFL students’ FLE. 

Among the positive psychology turn of emotion studies on second language acquisition (SLA) [12], and as a response to the concern that it is not enough to use only cross-sectional design to study FLE [7,13,14], the present research sets out to explore the role that CL plays in shaping FLE in an online learning context. Language learning has long been perceived as challenging [15]. When shifting from traditional classroom education to online distance education, in order to realize high-quality learning, primary concern should be placed on how to make the best of technology so as to guarantee learner–teacher and learner–learner interaction, thus transforming the challenging or commonly conceived disadvantageous situation [16] into a positive and fruitful one [8,17] in the context of SLA. 

In this regard, the present mixed-method study aims to explore, first, whether CL can shape FLE in an online learning context, and second, how self-appraisal of and reflection on the online cooperation experience made by each team can shape team members’ FLE and their perception of foreign language (FL) learning. In order to seek answers to these questions, a shortened-version FLE questionnaire, adopted from the one developed by Dewaele and MacIntyre in 2014 [8], was used to measure students’ enjoyment of online English learning. In the meantime, students from the experimental group were invited to complete CL activities including online presentation, teaching video creation, audiobook plot summary and comments, and to self-assess their CL performance. Their comments on CL experience serve as the qualitative data of the study. 

The current study may be useful for both university EFL teachers and learners, especially those who would like to make the best of online or blended learning activities. When online teaching and learning become necessary and even essential, either under emergency circumstances [18] or in a technology-assisted language learning (TALL) context [19], this study might serve as a timely response to the concern about how teachers can encourage active students’ participation so as to achieve positive language education [20,21] and boost students’ mental health and wellness during online learning. 

In the following sections, a review of relevant scholarship on emotions and FLE in second language education, CL in online and offline learning modes, and self-appraisal and self-reflection in an FL context is conducted, before moving on to present the research questions of this study. 

### 1.1. Literature Review

Emotions in SLA: Various emotions in FL teaching and learning contexts and individuals such as teachers and students who experience these emotions in various cultures and under different circumstances have been investigated. Among the negative emotions, boredom has arguably been regarded as one of the most thoroughly investigated emotions, both as a fabric of educational psychology [22] and as an element shaping FL pedagogy [23]. Much academic attention has been paid to learners’ boredom in specific FL contexts [24] and to how students’ FLB was shaped by teachers’ behavior and pedagogical strategies [2,25]. After the outbreak of COVID-19, substantial academic attention on boredom was shifted to students’ experience of such negative emotion during online classes [22]. It has been revealed that FL online class may be permeated with boredom if inappropriate interaction, be it underchallenging or overchallenging, has been initiated [26], and seemingly, both teachers and students involved in such a lack-of-interaction environment find it rather helpless when trying to figure out a method to improve the quality of teaching and learning [27]. 

As emotions such as FLB, FLE, and foreign language classroom anxiety (FLCA) probably coexist in educational contexts, scholars found that these three FL emotions may either positively or negatively predict one another. In particular, students who endure less FLB and FLCA are more likely to enjoy themselves in FL classrooms [2]. What cannot be ignored is that over one-third of Chinese students may experience FLCA [28], and this negative emotion has attracted the attention of many Chinese scholars who focused specifically on the anxiety experienced by students when they practice speaking [29] and writing [30] skills when attending offline classes. It has been discovered that both teacher-related and student-related factors may result in an anxious attitude towards FL learning. Moreover, based on the comparison between Chinese EFL learners and their counterparts in other parts of the world, it has been proven that the former are more likely to feel anxious because they feel embarrassed when making mistakes [31], and in the class of a college English course in China, the number of enrolled students can be relatively large—a factor which may lead to the slow improvements of FL learners’ language proficiency and even cause them to experience FLCA [32]. On the other hand, research also showed that FL learning does not necessarily lead to anxiety [33], as students may appreciate teachers’ strictness—a stimulating factor which can help to unlock their potential and help them to obtain desirable learning results. Research has also been conducted to investigate strategies that students applied to cope with foreign language anxiety [34]. 

In terms of studies covering the topic of online FL learning, there is research exemplifying the similar emotions experienced by students participating in both online and offline classes [35]. Some investigations, despite being set in a virtual education context, offer few discussions on the specificities of online FL classes [33]. For the studies which do cast an insightful light on online learning emotions, especially those published after the outbreak of the pandemic, they manifested in their findings considerable differences between online and offline emotions experienced by EFL students [36], and the focus of these studies was predominantly placed on negative emotions. 

Positive emotions can co-occur with negative ones, and the two kinds of emotions do not necessarily bear a “seesaw” relation, which means when one emotion goes up, the other may not go down accordingly [37]. The examination of positive emotions can help to discover how participants involved in it can appreciate the meaning of their activities [1]. With the establishment of positive psychology as a specific research field, especially the holistic view it provides on studying human emotions [38], researchers, instead of focusing exclusively on negative emotions such as boredom and anxiety, have taken into consideration the driving forces of positive emotions in SLA, such as love [39], pride [40], and sense of achievement [41]. It has been found that the reasons for triggering the emotion of love may vary from person to person, as some students enjoy role-playing while others find it pleasurable to read English novels [39]. Just as the feeling of love may be aroused under specific circumstances by various reasons, scholars also further proved the complexity of positive emotions relating to SLA by revealing that students’ emotions of pride can be categorized into different levels, from none to hubris [40]. Additionally, it has been confirmed that a sense of achievement correlates positively with enjoyment [41], and achievement emotions usually go hand in hand with EFL learning achievements [6,42,43]. 

The attention to which various studies on FL emotions have been devoted after 2020 shifted substantially. This can be regarded as a natural and timely response to the influence of the COVID-19 outbreak which has placed remote learning and teaching into a principal and vital position. 

Students’ FLE in the context of in-person classes: Both teachers and students’ emotions and the factors which cause these emotions have been examined. In terms of in-person classes, through the examination of teacher-related variables, it has been revealed that teachers’ friendliness [38], their frequency of using FL [44], and their sensitivity to students’ emotional appeals [6] can strongly predict students’ FLE. From learners’ perspectives, especially considering the samples of Chinese EFL learners, students’ attitudes towards teachers, their language fluency [31] and self-perceived FL proficiency [45], their engagement in class activities [9], and their learning achievement [42] may determine whether they would have an enjoyable FL learning experience or not. Moreover, in a large-scale research study on FLE involving over 2000 Chinese university students, it was found that the emotion of enjoyment correlated with EFL learners’ willingness to communicate [14]. Furthermore, the achievement of FLE is closely related to students’ aspects of perception, which indicates that EFL learners, even when situated in the seemingly discouraging situations, can appreciate the positive significance of FL learning [46]. Except for teacher- and learner-related factors, the classroom environment which can be cultivated by all the participants is likely to have a predictive effect on students’ FLE [47]. In terms of domain-specific studies on FLE, one investigation specifically examined EFL learners’ enjoyment in a listening class, finding out that enjoyment might serve as an effective impetus to in-depth exploration of listening materials and the horizon broadening of the FL learners [48]. 

FLE in online classes and CL: Although current research on FL emotions aroused during online learning deals predominantly with negative ones, there are still a handful of studies examining the enjoyment experienced by online FL learners, and meaningful discussions have been had on the relationship between FLE and CL. Students reported that remote learning could ease their fear of making mistakes and even boost their self-efficacy [7]. More importantly, a few scholars have shifted their attention to online CL, especially how such cooperation is related to FLE. Online writing, for instance, when conducted in a collaborative way, can become an enjoyable experience according to an investigation examining over 300 Chinese university students [49]. Furthermore, on a more general scale, research also revealed how e-tandem language learning which emphasizes mutual help between students can enhance their FLE online [50], as well as how students may take advantage of emotion regulation to achieve enjoyment through cooperation [51]. Therefore, what cannot go unnoticed is the current scholarship on CL and FLE. Taking into consideration the fruitful academic achievements made on CL, especially in terms of how CL is positively related to students’ motivation [52] and achievement in various language skills [53,54,55,56,57], this study, by prioritizing CL in an online context, further investigates the methods of realizing effective online learning for university EFL students. 

Self-appraisal in an FL context: This is a research field enjoying a long history and finding its significance in current studies of EFL learners. Students’ self-appraisal, as an important self-reflective activity [58], is distinct from teacher assessments. For psychologists, self-appraisal also entails a behavior conducted on an individual level, rather than on a social level [59]. The topic in SLA has attracted continuous attention from researchers who tried to figure out the difference between the results of self-assessment and those of other measures of language proficiency [60]. Students were not only invited to self-assess their language proficiency but what they believe should serve as proper FL classroom activities [61]. As for this study, students were also invited to conduct self-appraisal. However, what they were required to assess was not individual performance. Instead, they needed to assess their CL performance, and accordingly, to comment on their CL experience. This is related to a self-reflection activity which has rarely been explored in an online FL and CL context. One exception is the research undertaken by Jin et al. [10], in which it used reminiscence, or a positive way of recalling past experiences, as a factor that helped students reduce their FL classroom anxiety. 

Based on the above discussion on emotions in SLA, it can be seen that various positive and negative emotions have been examined under different circumstances, both online and offline. FLE evidently has received academic attention worldwide, as there are fruitful findings on students’ emotions, enjoyment included, made relying on the investigations of language learners of all levels living on different continents [8,11,39,62,63,64,65,66]. However, although positive as well as negative emotions are inevitable and ubiquitous in an FL learning context, there are few studies, if any, which directly deal with improving students’ online learning experience through investigating the role of CL in shaping students’ FLE. The current research, based on real tasks set up for the EFL learners and their actual experience of attending an online English class, attempts to explore the relationship between FLE and CL in a remote learning environment, which will provide an insight into why and how CL may shape students’ FLE when they participate in online learning or the online sessions of a blended learning course. 

### 1.2. Research Questions

In order to further facilitate the efficiency and enjoyment of online FL learning, this study aims to answer the following questions by collecting both quantitative and qualitative data. 

Does CL help to shape university EFL learners’ FLE in an online learning context? How and why?Do self-appraisal of and reflection on the online cooperation experience shape team members’ FLE and their perception of foreign language (FL) learning?

## 2. Materials and Methods

This study adopted the convergent mixed methods approach [67] to explore how CL can shape students’ FLE in online classes. To this end, quantitative and qualitative data were collected separately and were analyzed in regard to the role of CL in shaping learners’ online FLE experience. Moreover, it used individual case research [12] to explore why students may experience positive or negative emotions during CL.

### 2.1. Participants and Context

The participants were 98 Chinese EFL university students aged between 18 and 20 who were in their first year of university studies and enrolled in the listening classes taught by the first author. As a course of text-book-driven nature may appear to be boring to students [68], let alone in an online context where reduced interaction and a feeling of isolation may occur to students [69], the teacher applied the student-centered method, encouraging these EFL learners to develop their comprehensive English skills by exploring various listening materials. They were requested to complete a series of tasks based on the materials provided. For example, they were invited to produce presentation videos, summarize and comment on the audiobooks they listened, and share their learning achievements via Tencent Meeting (teng xun hui yi), one of the dominant online meeting platforms in China. The major difference between the two groups is that, for the experimental group, CL served as a key element in the student-centered method, whereas students from the control group did not work together in teams when completing tasks.

In addition, the two groups were not identical in terms of English proficiency. All the participants took the examination of College English Test Band 4 (CET-4), one of the most prestigious, national English proficiency tests. They all passed the examination, but the mean value of their CET-4 scores indicated that the students from the control group had better English language skills than those from the experimental group. On the other hand, for students from the same group, their English proficiency levels were similar. Such difference provides a fertile ground for exploring whether FLE is directly related to English proficiency in an online learning context. The detailed information of participants is presented in Table 1. 

### 2.2. Instruments

A bilingual version of the FLE questionnaire was presented to the participants who were encouraged to read the English original when completing the survey. They were also informed that if any confusion was caused by reading the English version, they could then turn to the corresponding Chinese version for help. Two criteria were considered when customizing the FLE questionnaire used in this survey. First, all the dimensions selected were highly relevant to the context of this research, including teacher (e.g., “The teacher is supportive”), personal (e.g., “I don’t get bored”), and social factors (e.g., “We form a tight group”). Second, we made sure that for each dimension measured in the questionnaire, the items used for measuring the dimensions matched well with the overall characteristics of the learning content and tasks, and only those that appeared to be redundant were removed. For example, the 21-item version of the FLE scale [8] includes the item “I can be creative”, which may not be helpful to reveal CL’s shaping influence of FLE in the present study, so it was removed from the scale. It is worth mentioning that a 9-item version of FLE has been developed recently [70], but the items in this shorter FLE scale so far cannot possibly cover all the questions that this research would like to investigate. For example, this scale, developed and validated by Botes et al. [70], does not include the item “I don’t get bored”, which is included in the 21-item version. The item “I don’t get bored” actually is especially relevant to the current study, as boredom is an emotion often experienced by students during online learning. This is why a shortened version of the existing FLE scale developed by Dewaele and MacIntyre in 2014 [8] was used in the survey. 

The FLE scale: A shortened version of the FLE scale developed from the 21-item version initiated by Dewaele and MacIntyre [8] was used in the survey. The shortened version of the FLE scale was comprised of 10 items, covering enjoyment (e.g., “I enjoy it”), classroom climate (e.g., “It’s a positive environment”), collaboration (e.g., “We form a tight group”), sense of accomplishment (e.g., “In class, I feel proud of my accomplishments”), and teacher support (e.g., “The teacher is encouraging”). Participants were asked to rate their level of agreement on the descriptions of FLE on a five-point Likert scale ranging from 1 (“strongly disagree”) to 5 (“strongly agree”). In the present study, the FLE scale achieved Cronbach’s alpha values of 0.78 and 0.85 at Times 1 and 2, respectively, both of which show good reliability. 

### 2.3. Procedures

First, all the students from both control and experimental groups were invited to complete the FLE scale online for the first time. Second, the students from the experimental group were invited to the Chaoxing Online Class (chao xing xue xi tong) to build their own team of 2 to 4 students. They were then provided with instructions about the CL tasks of the semester. 

The CL tasks, as shown in Figure 1, consist of two parts. The first part is about a task that is relatively time-consuming and demanding, which required each team to provide an education video about the key points covered in the unit. The first author, who was also the lecturer of this listening class, communicated with each team in advance to clarify the sections that needed to be elucidated by team members. She encouraged each team not only to explain words, phrases, and collocations appearing in these sections but to explore the cultural elements relating to the content and to comment critically on the views expressed in the dialogues and by the lecturers. This means that CL in the study’s context is not one-dimensional. In other words, instead of focusing on one specific domain such as reading [71] and writing [72], this study, through prioritizing listening practice, aims to improve students’ overall English skills through CL-based tasks. For the second part, throughout the semester, students from the experimental group were required to listen to three audiobooks, *The Call of the Wild* (1903) by Jack London, *The Great Gatsby* (1925) by F. Scott Fitzgerald, and *Charlette’s Web* (1952) by Elwyn Brooks White. The reasons for choosing these novels were because they are so well-received that they are now regarded as classics of all time and because the audios of these books were well-made and suitable for students to enjoy while developing their listening skills. 

Third, after a 3-month intervention, both groups once again completed the FLE scale for the second time. In addition, all the students were invited to answer two open-ended questions, including first their overall impression about the course, and second, what suggestions they would like to make for the lecturer of the course. All the data were collected online, and all the assignments completed by students were emailed to the lecturer. 

For students in the control group, they received the ordinary instructions of the listening class, and they prepared presentations and listened to the audiobooks all by themselves, without any kind of CL activities involved. 

### 2.4. Data Analysis

Data collected were analyzed in three steps. First, the reliability of the FLE scale used was tested by Cronbach’s alpha, and a paired *t*-test was also performed to explore the changes in FLE over the pre- and post-test times between the experimental and control groups. Second, descriptive statistics about the participants were generated, including mean, standard deviation, and other relevant information. Third, a detailed text analysis on the reflective comments provided by teams from the experimental group was conducted. During the process, both authors picked up and summarized the main themes from each comment and figured out the similarities and differences between different teams’ CL experience. These initially identified themes were then re-categorized based on their shared common features, and comments with distinctive and unique features were also marked. 

## 3. Results

The results are reported in two sections, guided both by the research questions and the types of data collected.

### 3.1. CL and FLE in Online FL Learning

A paired sample *t*-test was performed in the between-survey FLE scale scores for each group. The results reveal that the change in the FLE scale was significant for the experimental group (*p* < 0.0001), whereas for the control group the change was not significant (*p* = 0.10). Other descriptive results, including changes in means and standard deviation (SD), are presented in Table 2. 

In addition, the comparisons between the two times of FLE survey are presented in Figure 2 and Figure 3. They demonstrate, respectively, the changes in FLE rated by the two groups at Time 1 and Time 2. As can be seen from the two figures, the overall FLE of the experimental group enhanced significantly, whereas that of the control group fluctuated dynamically. 

### 3.2. Qualitative Data on Cooperation and FLE

According to the reflective comments on CL provided by team members of the experimental group, which was provided when they completed the chart presented in Table A1 and Table A2 of Appendix A, four major discoveries were made, the key points of which are illustrated in Figure 4. First, whether the assigned task can be completed by responsible and cooperative teammates determines the FLE of the students. Meeting challenges together and solving problems along the way so as to complete a satisfying task was found to be the sole and ultimate goal for the teams whose self-appraisal scores were high, as students from Team 3 recalled:
*It took a long time to prepare, but we appreciated the experience as the work we produced was satisfactory. The night before the presentation, we conducted a trial to familiarize ourselves with the procedure of using Tencent Meeting—trying our best to ensure that presenters would not have to face any technical problems. In addition, the division of labor was clear, and cooperation was efficient. Whenever a problem was raised, everyone would work together to find a solution, and what is more, we would contact the teacher in time if needed. Cooperation enabled us to achieve the best results. Thanks to CL, while preparing the presentation, we thoroughly studied the content of the original listening text. We not only acquired new knowledge, but learned a lot from each other.*

Second, as articulated in the above comments, whether new knowledge and new skills can be acquired is regarded by team members as a determining factor that can shape their FLE in online English classes. On top of that, improvements also include the ability of using new software tools, as students from Team 9 and Team 10 commented:
*The experience of cooperation was very pleasant. When we prepare the task together, we learned a lot of new knowledge and exchanged our views. Overall, the cooperation went smoothly*. (Team 9) 
*The three of us took turns to complete different parts, so that each of us had the opportunity to develop different language skills. We not only read wonderful and interesting novels, but also improved our skill of summarizing. It is more than enjoyable to complete the homework with friends. Besides, through cooperation, we learned a lot of grammatical knowledge and new expressions*. (Team 10)

Third, as students may endure various stressful situations during online learning, CL plays an active role in helping them relieve the pressure. It can be challenging when learners have to complete their tasks in their own dwelling places instead of in classrooms surrounded by peers. Moreover, the course’s student-centered nature and its task-oriented design may leave learners with the impression that the tasks assigned are demanding. This means that they need to use five language skills (reading, writing, speaking, listening, and translating), the abilities of background research, and the competence of using online learning platforms and related software to complete the assigned tasks, as students from Team 15 said:*Everyone quickly chose the parts each would like to work on. All of us participated in the production of the PowerPoint. During the video recording process, everyone was well prepared. Occasionally there were minor flaws, and we would record that part once again. In the end, we further edited the video. Everyone prepared the team assignments carefully, and when group members came across any problems or felt confused, we would work together to solve these problems. In other words, everyone helped each other and we cared about each other’s progress. That also explains why the experience of video recording and editing though appeared to be tiresome was actually pleasant and exciting. The outcome was perfect. Everyone worked efficiently and cooperated actively.*

Fourth, members of the teams may experience enjoyment when they find out they share similar feelings on certain issues. These include similar emotions experienced when listening to the audiobooks, watching and editing videos, and the sense of accomplishment aroused after making improvements. Students from Team 5, for example, shared their delightedness when reading the book *Charlette’s Web*: *It’s a very interesting cooperation! Even in the midst of the epidemic, the colorful contents of the book enable us to enjoy literature and experience the joy of reading together.*

It should be noted that how satisfied students feel about cooperation may influence the outcome of their assignments and then further determine the score of their team’s self-appraisal. For the teams who reported a pleasant and rewarding experience of cooperation, their reflective comments were mostly positive, and even when they recalled regrets concerning certain assignments, they still could see them from the bright side. In other words, they would tend to believe that in the future, when coming across similar tasks, they would definitely be better prepared and more experienced, thus producing more satisfying work. 

On the other hand, for some team members, no matter how good each individual’s command of English was, a lack of cooperation was directly related to a low sense of accomplishment, and consequently, a low–intermediate score might be given when assessing their own CL performance. The self-appraisal score of each team from the experimental group is illustrated in Figure 5, and a detailed summary of the comments from each group is presented in Table A3 of Appendix A. 

## 4. Discussion

The main findings of this study, which focuses on CL’s role in shaping students’ FLE, are twofold and can be categorized into two interdependent groups underlining internal and external factors, respectively. First, internal factors entail students’ motivation, peer support and interdependence, and self-reflection. This study confirms that positive emotions are closely connected with students’ motivation of learning [62] and their eagerness for achieving great improvements. One feedback received from Team 5 of the experimental group said that their cooperation was so pleasant that they even forgot time when working on the education video. This motivation, as this study reveals, does not necessarily or directly relate to students’ FL abilities. Although it is possible that the higher the FL proficiency one enjoys, the more joyful he or she might be during FL learning [73], this study reveals an alternative situation when peer support and interdependence also play a key role in enhancing FLE. 

This is because different team members, though their language skills may vary, can make the best out of CL through positive interdependence. Our findings are confirmed by the feedback of Team 3, Team 10, and Team 15 of the experimental group, who reported that mutual improvements were guaranteed as they learned from each other and met challenges together in order to present high-quality teamwork. Moreover, good teamwork can win students bonus points that cannot be gained otherwise when working alone. This is because, during the CL process, team members’ strengths could complement each other’s deficiencies. As members of Team 5, Team 10, and Team 11 recalled, they highly appreciated the modification, correction, editing, comments, and suggestions provided by their team members, which not only helped them to improve their abilities but to a great extent boosted their enjoyment. Just as students who receive good marks in a test may feel proud [40], feedback from team 18 also highlights that it is useful to take charge of different parts of the tasks so as to achieve general and domain-specific English improvement. 

Good CL efforts therefore can help students to experience enjoyment, especially when good grades have been achieved and various abilities including language skills have been improved. As revealed in the marking criteria of self-appraisal (see Table A1 in Appendix A), which was modified based on the factors of positive goals and source interdependence, which are aspects designed to measure classroom life [74], students who enjoyed CL were mostly confident, provided satisfactory scores of self-appraisal, and would feel more capable in dealing with similar tasks assigned by other lecturers in the future. Even more importantly, as students from Team 10 commented, thanks to the CL experience, they could confidently predict that in the future they would have even better performance. In other words, such an optimistic attitude may help guarantee greater enjoyment in future FL learning, as they can, through such cooperation experience, take full advantage of mutual support and personal as well as academic interdependence. This discovery is in line with the findings that personal peer support (PPS) and positive goal interdependence (PGI), two factors relating closely to CL, can boost FLE [13]. This is particularly true in an online learning context, where, according to research on FL learning in a virtual environment, reduced interaction has been observed, compared with classroom learning [36], and a feeling of boredom becomes dominant [27]. 

More specifically speaking, in terms of dealing with boredom in online English language classes, CL can be a recommendable solution, as students may consciously attempt to deal with negative emotions felt during online learning [27], and they tend to be affected by the emotions and the degree of motivation of their peers [24]. Mutual support concerning language improvement [50], knowledge exploration, and solutions to technological problems encompassed in this study can serve as useful references for teachers who would like to help increase EFL learners’ online FLE. Furthermore, although studies about online FL learning usually list reduced interaction and increased boredom [7] as inevitable and often undesirable disadvantages [65,75,76], this study discovered that peer interaction is facilitated by CL, and peer presentations during which the presenters invite classmates to answer questions and share opinions proves to be an effective way of enhancing students’ FLE in an online class. This confirms previous discoveries that a positive online learning climate can be shaped by interactions, and at the same time, students’ social and emotional enjoyment within their group can also shape their FLE [49]. In this regard, if proper CL activities can be organized with adequate teacher and peer support, online teaching in the future should not merely be regarded as a forced alternative, such as in the situation faced by many FL teachers and students during the pandemic, but an effective tool where students can best transcend the barrier of space and make the most out of the FL classes through cooperation. 

Another notable part of our findings is that although it has been confirmed that English proficiency can shape students’ FLE [1], this study revealed that in an online learning context, good language proficiency might not be directly related to enjoyable English learning experience. This is not only manifested by the results between control and experimental groups but also by the comment from Team 20 of the experimental group. According to the reflective comments of Team 20, each individual believed that they had a good command of English but, nevertheless, their self-appraisal score was relatively low, as there was little cooperation and, consequently, little sense of achievement or enjoyment was gained. This shows that, first, in line with the findings of previous research [9], FLE can be context-sensitive and may change dramatically. Second, considering the physical and mental boredom which might overwhelm students during online learning [69], those who worked alone and were faced with both emotional and technological challenges by themselves probably found it difficult to cope with all the issues at once. The lack of communication as well as personal interaction therefore can make them feel isolated, a finding which has also been proved by previous research [77]. Third, for intermediate and advanced English learners, if proper CL strategies can be implemented, it may be possible to achieve desirable English learning results. This is also confirmed by students’ responses to the two open-ended questions, as those from the experimental group commented positively regarding their FL learning and CL experiences, demonstrating remarkable enjoyment experienced in the online course. On the contrary, the responses from the control group appeared to be less optimistic, and a sense of boredom and anxiety could be traced as students expressed their concern about little interaction and the challenges of completing various tasks all by oneself. 

The study also revealed that, closely linked to self-appraisal, the contents of team reflection entailed both positive and negative elements, but positive reminiscing dominated the reflective comments. This means that, first, when recalling CL experience. students tend to remember the pleasant experiences as long as their cooperation is satisfying and fruitful, and second, the reminiscence process can stimulate students to consciously seek out solutions to the problems that might trigger negative emotions, thus creating a more positive attitude toward FL learning. This discovery is in line with the research on the reminiscing process of EFL learners [10], as when students look back on the setbacks and challenges of FL learning, they may not be overwhelmed by the negative emotions caused by these unpleasant memories. Instead, students may place emphasis on positive reappraisal [78], which, along with resilience, can boost students enjoyment. The finding also confirms the results from previous discoveries that a high level of FLE can positively correlate with students’ confidence [45], and in the case of this study, great FLE may positively predict teams’ self-appraisal scores. There is evidence found in the comments from Team 11 of the experimental group. The students from Team 11 believed that although revising and modifying their assignments called for painstaking efforts, such time-consuming work could nevertheless trigger enjoyment, as great improvements in terms of language skills, cooperative abilities, and technological skills were also made thank to this experience. Members of Team 11, accordingly, gave themselves a full mark in self-assessment, indicating their satisfaction with CL and their enjoyment of FL learning. 

What is unique about this study is that, first, students of the control and experimental groups were enrolled in a real teaching context, and second, the English proficiency of the two groups was not identical. The first uniqueness highlights that the participants under examination are not invited and instructed for experimental purposes only. Instead, they were fully involved in their learning activities and experienced different learning modes of a listening class at the same time. This can be different from studies in which the data collected were provided from participates who were specifically invited to join an FLE study. This also confirms that it can be inspiring and useful to study emotions of SLA that emerge when people use languages in real cases and in real time [79]. Second, although the students from the control group presented higher English proficiency compared with their experimental group counterparts, it does not necessarily mean that one’s English ability is directly linked to the enjoyment they experience in an FL class. There are various studies which provide insight into the relationship between language skills and FLE [14,43,80], and the present research may serve as an insightful addition, revealing that positive emotions can be achieved relying on good personal and academic interdependence. 

External elements include teacher support and technological skills, which form a close interdependent relationship with the internal factors discussed above. In other words, being external does not mean less important, and in the context of this study, it refers to the exterior factors that connect closely with students’ CL. They are not initiated by students, such as teacher support, and are not directly related to language acquisition, such as students’ and teachers’ abilities to master software tools. Yet, they both play significant roles in shaping students’ FLE, as they also entail various aspects of cooperation in an online learning context. 

As for the topic of teacher support, it has enjoyed constant attention in the past decade, and the key points made in relevant studies revealed that whether students can experience positive emotions or not depends crucially on teachers’ friendliness and [38] support [39]. This study provides new evidence for this view. In order to increase FLE, teacher support should be constant, positive, and multidimensional, providing both various well-chosen activities and individualized instructions for learners [14]. 

On top of that, the present research also found that if teachers can truly position themselves as equals to students, they may stimulate students’ motivation for further proceeding with their CL activities. The reason why Team 3 and Team 18 regarded the teacher of the listening class as a part of their group is that the teacher positioned herself as an ordinary team member and tried to provide inspiration instead of fixed answers to the questions students raised. This kind of teacher support serves as an additional contribution to guarantee the quality of CL. In this study, such support has proven to be fruitful, as it assisted students to fulfill their true potential of cooperation. Students were encouraged to explore cultural elements and to offer their unique and critical opinions on various topics covered in different sections of the listening textbook. 

In terms of technological skills, students expressed that the “extra bonus” brought by online learning was encouraging, as they started to learn how to use software tools to guarantee high-quality work production. This also echoes the discovery made from a comparative study on emergency remote teaching and in-person classes [7], when students engaging in remote learning can develop new strategies to cope with the situation. Such development and improvement of technological skills may also prove to be useful in the future, as students can feel more confident in presenting their ideas clearly with the assistance of multimedia forms of communication. 

## 5. Implications and Limitations

### 5.1. Theoretical and Practical Implications

The current study, through exploring CL in an online learning context, enriches the significance of FLE enhancement measure by highlighting how multidimensional, personal interdependence may positively shape learners’ FLE. Such interdependence entails academic mutual support, such as grammatical error correction and peer review of the writing tasks, as well as technological mutual support, as students may try together to solve problems of presentation creation using PowerPoint, video recording, and video editing. Both academic and technological support may contribute to the boost of FLE, as students may experience positive emotions such as a sense of accomplishment and achievement after completing a challenging task together. Furthermore, this indicates that self-appraisal is not only applicable on an individual level but can turn out to be a useful tool, stimulating students to reminisce on their CL performance and further help them to increase their FLE.

The findings of the present research may serve as helpful reference to both university EFL teachers and students, especially those who will organize or participate in online teaching or learning activities. Teachers should design CL activities that are suitable and perhaps “unpredictable”, leaving students the impression that the FL course is not simply textbook-centered but provides fertile grounds for them to develop comprehensive language skills and cooperation abilities through CL. Teachers may also encourage students to assess their own performance, and at the same time, invite them to comment on their CL experience, during which these EFL learners may, through self-reflection, figure out their advantages and shortcomings. More importantly, they can, through recalling these CL activities, recognize ways to achieve further improvements. Students, similarly, can reach out to both peers and teachers for support during online learning. The findings of this study can shed insightful light on how students can adjust themselves and make the best of online learning, as well as on how they can manage to enjoy good mental health in a remote learning environment. 

### 5.2. Limitations

The present study presents a helpful attempt of using self-appraisal comments as qualitative data to perceive EFL learners’ feeling about CL in online English classes. Future studies may consider using semi-structured interviews so that more case-specific information on students’ FLE and their CL experience can be collected and analyzed. Additionally, this research focuses exclusively on an online learning context. Future studies may adopt a comparative point of view, studying the role of CL in both online, offline, and blended learning contexts, so as to figure out the shaping factors of FLE under various circumstances. Lastly, the study was conducted in a real teaching context, which in the given situation means that the researchers could not possibly invite participants of both control and experimental groups with almost identical English proficiency. Studies in the future might consider inviting students with almost identical English proficiency to further investigate whether in that situation CL can shape FL learners’ FLE during online learning. 

## 6. Conclusions

This research study is among the pioneering ones which demonstrates why and how CL may shape EFL students’ online FLE, and why self-appraisal of CL performance and reflection about CL experience can positively predict FLE during online learning. The findings of the current study may be helpful for both teachers and students, who, in different situations, may be engaged in online teaching and learning activities. In particular, the current research serves as a useful reference, elucidating how students’ mental health can be improved in a negative-emotion-dominated online learning context. As online and blended learning become increasingly important and regarded as efficient and even crucial ways of education, the suggestions made in the study may help to boost students’ positive emotions during online learning and help to cultivate happier and more confident FL learners equipped with language, cooperative, and technological skills. 

## Figures and Tables

**Figure 1 ijerph-20-00611-f001:**
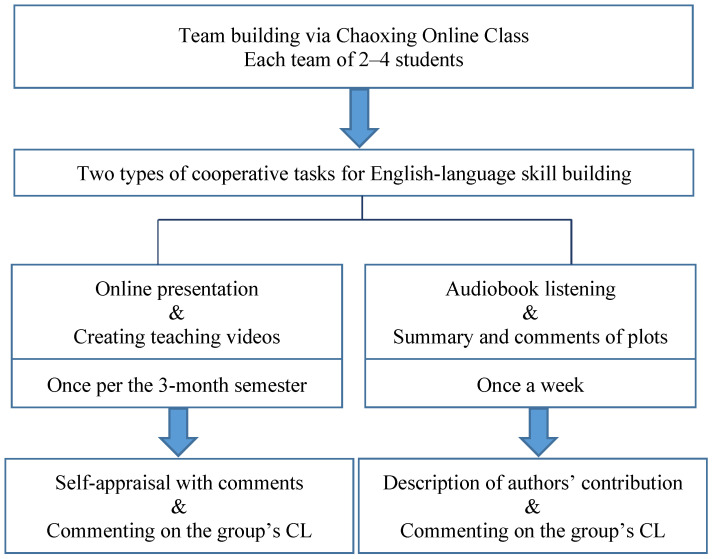
Students’ CL activities, self-reappraisal, and reflective comments.

**Figure 2 ijerph-20-00611-f002:**
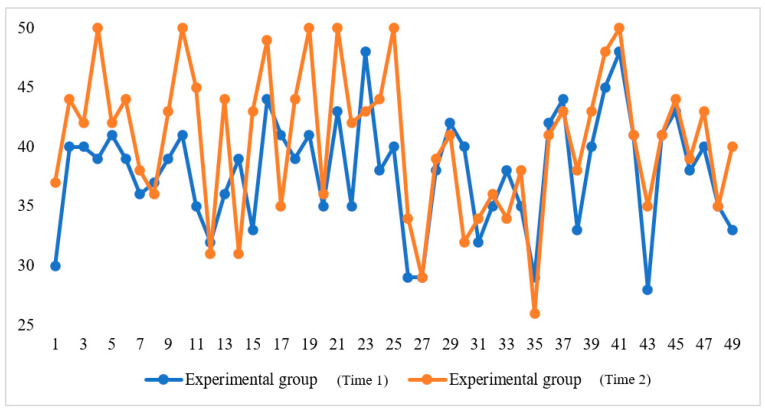
The changes in students’ FLE (the experimental group).

**Figure 3 ijerph-20-00611-f003:**
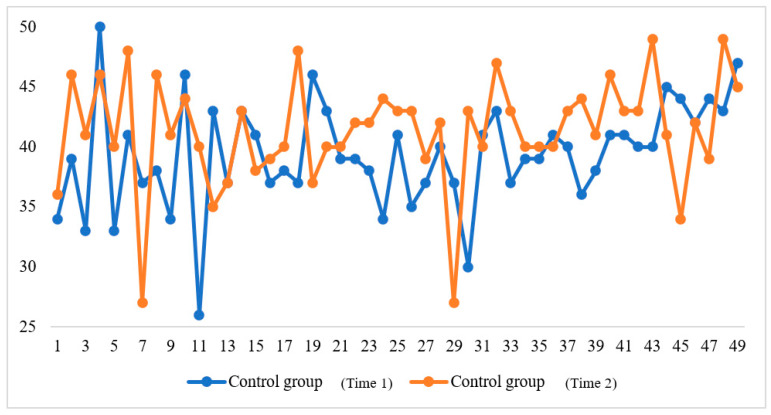
The changes in students’ FLE (the control group).

**Figure 4 ijerph-20-00611-f004:**
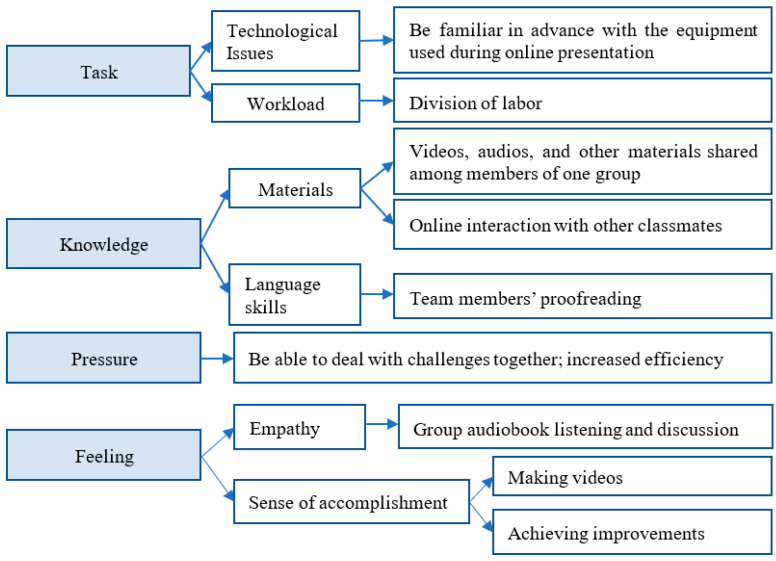
Factors boosting EFL learners’ FLE during online cooperative learning.

**Figure 5 ijerph-20-00611-f005:**
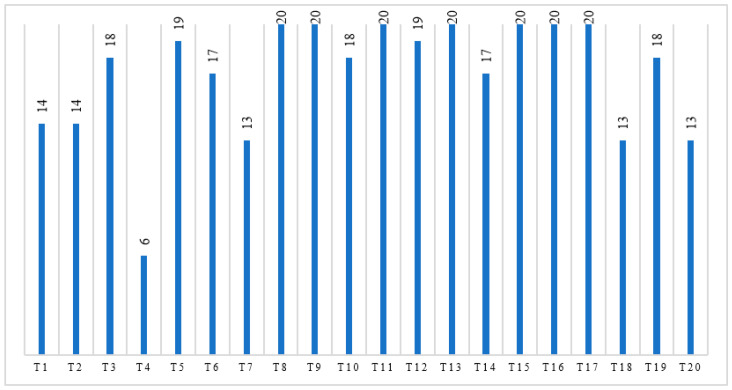
Team self-appraisal scores (the experimental group; “T” stands for “Team”).

**Table 1 ijerph-20-00611-t001:** Background information of the participants.

	Experimental Group	Control Group
Age		
Mean	18.61	18.78
SD	0.67	0.59
Minimum	18	18
Maximum	20	20
Years of learning English		
Mean	11.34	12.96
SD	2.34	1.65
Minimum	7	10
Maximum	15	17
Gender		
Male	10	29
Female	39	20
CET-4		
Mean	541.08	580.20
SD	58.58	48.94
Minimum	404	425
Maximum	643	688

**Table 2 ijerph-20-00611-t002:** The descriptive results of the FLE scale.

Scale	Times	Groups	Minimum Score	Maximum Score	Mean	SD
FLE	Time 1	Experimental	28	48	37.94	4.79
		Control	30	49	40.22	3.91
	Time 2	Experimental	26	50	40.55	5.98
		Control	27	49	41.78	4.19

## Data Availability

The data presented in this study are available on request from the corresponding author.

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
