# Peer review of "Enhancing Foreign Language Enjoyment through Online Cooperative Learning: A Longitudinal Study of EFL Learners"

_ijerph, 2022, doi:10.3390/ijerph20010611_

Round 1

Reviewer 1 Report

Review of ijerph-2074654- Enhancing Foreign Language Enjoyment through Online Cooperative Learning: A Longitudinal Study of EFL Learners

I have read through the paper with a strong intention to recommend its publication. The study reported in this paper focuses on university students’ foreign language enjoyment (FLE) in an online cooperative learning (CL) environment. The authors took a positive psychology approach to the examination of how and why CL may shape FLE.  A total of 98 Chinese university students of English as a foreign language (EFL) were assigned into experimental (n=49) and control groups (n=49). Both groups completed a short-form foreign language enjoyment (FLE) scale before and after a 3-month intervention. As the authors report, “the students in the experimental group were assigned with tasks that needed to be accomplished by teamwork. Moreover, each team was also requested to reflect upon their cooperation experiences and self-assess their performance of these tasks.” The results showed that the overall FLE of the experimental group increased remarkably, whereas that of the control group fluctuated considerably. Furthermore, analyses of experimental group students’ self-appraisal comments revealed that students with pleasant cooperation experiences usually experience high FLE, give satisfactory marks on their performance and cast a positive light on their future learning and development. The findings and implications provide meaningful insights in terms of how to boost online FLE through CL so as to promote positive mental health of students in a technology-assisted language learning (TALL) context.

I have two queries and/or recommendations. The first concerns the methods for data collection. In section 2 Materials and Methods, the authors state:  “This study adopted a mixed-method research design to explore how CL can shape students’ FLE in online classes, and moreover, it used individual case research [12] to explore the reasons why students may have positive or negative emotions in CL.” Creswell (2015) talks about three ways of mixed-methods designs in educational research. Can the authors be specific on their way of “mixing” by using clearer labels when they talk about this?

The second point is about the authors’ reference to the existing literature. Given the prominence of notions such as enjoyment, boredom and anxiety, I would rather recommend that they consult some studies on Chinese EFL/ESL learners' anxiety to ground their study well.  The authors did a good job in referring to a good array of recent studies in the areas of enjoyment and boredom, but not on anxiety. Two papers come in my mind instantly, which are clearly about Chinese learners of English, who were investigated for the role of anxiety. The authors might find them useful.

Yu Y and Zhou D (2022) Understanding Chinese EFL learners’ anxiety in second language writing for the sustainable development of writing skills. Front. Psychol. 13:1010010. https://doi.org/10.3389/fpsyg.2022.1010010

Liu, M. (2006). Anxiety in Chinese EFL students at different proficiency levels. System, 34(3), 301-316. https://doi.org/10.1016/j.system.2006.04.004

I also ask the authors to double check their use of English. In general, the paper is clearly written. But there are places where some expressions are a bit awkward. See below for some of the instances:

Repetition below:

Amongst the positive psychology turn of emotion studies in a second language acquisition (SLA) context [12], and as a response to the concern that it is not enough to use only cross-sectional design to study FLE [7, 13, 14], the present research is set out to explore the role that CL plays in shaping FLE in an online learning context.

The word “context” does not have to appear twice in the same sentence.  The first “context” can be deleted:

Amongst the positive psychology turn of emotion studies in a second language acquisition (SLA) [12], and as a response to the concern that it is not enough to use only cross-sectional design to study FLE [7, 13, 14], the present research is set out to explore the role that CL plays in shaping FLE in an online learning context.

Misuse of words, for example:

much of the attention on [sic, to] foreign language boredom (FLB)

… attention to …

In regard to Chinese students’ anxiety, especially based on the comparison between Chinese EFL learners and their counterparts in other parts of the world, it has been proved that the former are more likely to feel anxious because of the convention of education and the cultural factors that may cause them feel embarrassed when making mistakes [28].

More citations are needed here to support the authors’ argument.

Also, … has been proved… should be “… proven”…

Reviewer 2 Report

Dear authors,

Thanky ou for the opportunity to read your article. I find the issue of cooperative learning important and especially, when reducing student´s anxiety and thus improving their language skills. I consider your article one of the best  I have read in recent years. I would just recommend its proofreading by a native speaker. Overall, well-done!

Best,

Reviewer

Reviewer 3 Report

The study is a quantitative and qualitative analysis of foreign language enjoyment in an online cooperative learning, which is an important and current topic. Main advantages:

It is a longitudinal study, which corresponds to the need for this type of research often indicated in the literature

The rarely analysed emotion of joy is explored.

Current literature review.

What could be improved:

The authors could place their operationalization of the analysed concept at the beginning of the article.

Varying levels of proficiency of the participants should be listed as a limitation as it could have affected the score.

The relationship between students' adjustment/ mental health and the study results could be more emphasised, especially in the conclusions, as it is the leading topis of this issue. 
